# Modelling peptide–protein complexes: docking, simulations and machine learning

Arup Mondal[1,2] ⬥, Liwei Chang[1,2] ⬥ and Alberto Perez[1,2]* ⬥

[1]Department of Chemistry, University of Florida, Gainesville, FL 32611, USA and [2]Quantum Theory project, University of Florida, Gainesville, FL 32611, USA

## Research Article

**Key words:**
Docking; force field; peptide binding; peptide–protein interaction; machine learning; molecular dynamics simulation; scoring

**Author for correspondence:**
*Alberto Perez,
E-mail: perez@chem.ufl.edu

A.M. and L.C. contributed equally to this work.

## Abstract

Peptides mediate up to 40% of protein interactions, their high specificity and ability to bind in places where small molecules cannot make them potential drug candidates. However, predicting peptide–protein complexes remains more challenging than protein–protein or protein–small molecule interactions, in part due to the high flexibility peptides have. In this review, we look at the advances in docking, molecular simulations and machine learning to tackle problems related to peptides such as predicting structures, binding affinities or even kinetics. We specifically focus on explaining the number of docking programmes and force fields used in molecular simulations, so a prospective user can have an educated guess as to why choose one modelling tool or another to address their scientific questions.

## Introduction

Protein–protein interactions (PPIs) are a vital component of pathways regulating the behaviour of cells. In disease, some of these pathways become aberrant; therefore, identifying ways of inhibiting them is of great therapeutic importance. Inhibiting PPI with small molecules is not always possible due to the large interface region, lack of binding cavities and specificity of the interaction. However, between 15 and 40% of PPI (London *et al.*, 2013) are mediated by peptide epitopes, giving rise to opportunities for peptide-based inhibition of PPIs. There is a growing market for peptide-based therapeutic agents (Martins *et al.*, 2021; Wang *et al.*, 2022), and there are already more than 60 peptide drugs approved in the United States (Usmani *et al.*, 2017; Lau and Dunn, 2018). Peptides have well-known degradation pathways, lower toxicity than small molecules and are highly specific. Some of the challenges for peptide-based therapeutics such as the rapid degradation by proteases or limited ability to cross membranes (Fosgerau and Hoffmann, 2015) can be overcome by using modified amino acids and cyclisation techniques (Bechtler and Lamers, 2021). Others, like efficient delivery strategies (e.g., oral delivery), limit broader interest (Ganesh *et al.*, 2021).

The rational design of peptide-based therapeutic requires structural knowledge of peptide–protein complexes at the atomistic level. Experimental studies are challenging and expensive as peptide degrades fast, and the spectrum of target candidate is too broad to explore experimentally. While the use of computational pipelines is well-stablished in the early stages of drug discovery for small molecules, peptides present some unique challenges that have limited the success of computational pipelines for inhibitory peptide design. Structurally, peptides interact with proteins in different ways (Arkin *et al.*, 2014): 1) as coils through specific amino acid interaction; 2) by adopting well-defined secondary structures (e.g. hairpins or helices) and 3) through discontinuous interactions along the peptide chain. In these interactions, peptide flexibility is important, as they are often intrinsically disordered in their free form and adopt well-defined structures upon binding – unlike small molecules where flexibility is more limited.

The reader is refered to several excellent and exhaustive reviews on peptides as therapeutics, and the role of docking to identify PPIs (Fosgerau and Hoffmann, 2015; Ciemny *et al.*, 2018; Apostolopoulos *et al.*, 2021). In this review, we describe three major classes of computational methods that are routinely used to elucidate different aspects of PPIs: 1) docking; 2) molecular dynamics (MD) simulations and 3) machine learning approach. Docking approaches have traditionally been the most successful at exploring the possible orientations and interaction sites between proteins and peptides. Their use for peptide systems has evolved from protein–small molecule and protein–protein docking tools and face different challenges when accounting for the highly flexible nature of peptides. Hence, these methods are described in most detail. MD methods draw from the wealth of enhanced simulation methods available in the literature. Their purpose is to confer detail either about the binding energy landscape or binding mechanism. They are generally not high-throughput methods and complement studies where the bound structure is known. Finally, machine learning approaches are rapidly evolving, thanks to Alpha-Fold's recent success in protein structure prediction. These methods have rapidly been ported to

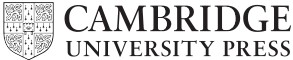

other applications such as peptide–protein structure prediction with a success rate to match those of the best performing docking programmes. Although different in methodology, the three classes of methods need to account for the challenges in sampling bound conformations as well as identifying them through some scoring function. Our goal with this review is to identify the current strategies to approach these challenges and provide a broad understanding of the advances and limitations in the field.

## Docking section

Docking remains an efficient approach to sample bound conformations given the receptor structure and the ligand. Their success in small molecule-protein docking and their ease-of-use through webservers and standalone software have popularised this method for virtual screening in the early stages of drug discovery (Taylor *et al.*, 2002). Some of this success has been translated into the protein–protein docking field, as seen from the evolution of predictions in the CAPRI (Critical Assessment of PRediction of Interaction) competition (Lensink *et al.*, 2017, 2020). Despite these successes, peptide–protein docking remains a more challenging problem (Ciemny *et al.*, 2018). The success of docking relies on the ability to sample bound conformations, and the ability to identify native-like poses using a scoring function. The flexible nature of the peptides significantly increases the sampling problem with respect to small molecule docking and limits sampling native-like poses (Rentzsch and Renard, 2015). Similarly, the flexibility challenges the ability to transfer standard scoring functions to identify peptide–protein complexes. For example, despite the poly-aminoacidic nature of peptides, a straightforward application of protein scoring functions has limited success. Thus, modifications to protein–protein scoring functions are needed to correctly identify native-like poses (Agrawal *et al.*, 2019; Weng *et al.*, 2020). Innovation in the peptide docking field comes from strategies for more efficient handling of flexibility and overcoming limitations in scoring. Along with these improvements, the curation of peptide–protein databases is crucial for systematic testing, benchmarking and assessing these docking methods (Hauser and Windshügel, 2016). There are already excellent reviews and benchmark studies assessing the performance of different methods (Wang *et al.*, 2016; Ciemny *et al.*, 2018; Agrawal *et al.*, 2019; A. C.-L. Lee *et al.*, 2019; Weng *et al.*, 2020). Hence, we will limit this section to the nature of the databases and different sampling/scoring strategies prevalent in the field.

## Databases

Many peptide docking methods have evolved from either protein–protein or protein–small molecule docking tools. Their modification includes better handling of flexibility and specific scoring functions. To test their performance on peptide–protein complexes, several efforts distil the structural information from the PDB, identifying sets of peptide–protein complexes amongst the ~150,000 structures deposited in the RCSB-PDB (Berman *et al.*, 2002). Thus, the emergence of databases for peptide–protein complexes streamlines the process and accelerates the advancement of the field. Several databases are available, each with a specific purpose in mind – ranging from properties such as length of the peptides to types of binding motifs. Here, we provide an overview of the widely used databases. Table 1 provides a quick reference summary.

*LEADS-PEP* (Hauser and Windshügel, 2016) consists of 53 peptide–protein complexes with peptides ranging from 3 to 12 amino acids with resolution lower than 2 Å. The entries originate from a clustering on sequence space, retaining only complexes that are diverse in terms of sequence – excluding those that interact with DNA or RNA. Due to the short length of the peptides in this database, it is suitable for benchmarking docking tools originating from small molecule docking programmes.

*PeptiDB* (London *et al.*, 2010) is a non-redundant database of 103 high resolution peptide–protein complexes. These peptides are 5–15 amino acids long with diverse bound conformations (helix, β-strand and coil) and functionalities (such as signal-transduction, antibody binding, protein trafficking and transporting). The set includes complexes with a significant conformational change upon binding. These characteristics make PeptiDB appropriate for benchmarking docking tools that account for peptide flexibility.

*PPDbench (*Agrawal *et al.*, 2019) database has been used to benchmark 6 common docking programmes which contains 133 peptide–protein complexes with less than 40% sequence similarity. The set is diverse with respect to functionality, but the range of peptide lengths is narrower (9–15 amino acids). The benchmark study with this database by Agarwal *et al.* showed that different docking methods perform best on different class of peptides, classified in terms of their functionality such as enzymatic, signalling and many others (Agrawal *et al.*, 2019). In another study, Weng *et al.* created and used the *PepSet* (Weng *et al.*, 2020) database to benchmark 14 docking programmes. This database contains 185 high resolution peptide–protein complexes with less than 30% sequence similarity and peptide lengths ranging from 5 to 20 amino acids.

Both LEADS-PEP and PeptiDB have been widely used for benchmarking sampling and scoring ability, but they are limited by peptide length. Peptides longer than 20 amino acids are very common in nature. The *PepPro (*Xu and Zou, 2020) database contains 89 non-redundant peptide–protein complexes with longer peptide sequences (5–30 resides) and diverse peptide secondary structures. As a useful feature of PepPro, the database contains 58 structures of the unbound receptor proteins – making it ideal to benchmark docking methods for predicting *apo* to *holo* conformational changes.

While the above databases have a limited number of complexes, a few databases are more inclusive in their search parameters. For example, *PepX* (Vanhee *et al.*, 2010) contains 1431 non-redundant complexes from the PDB with peptide size ranging from 5 to 35 amino acids and resolution less than 2.5 Å. There are redundancies in the database and the number of complexes can be reduced by clustering on their interaction interface reduces, resulting to 505 unique cluster centres. Similarly, the *PepBind (*Das *et al.*, 2013) database is built on similar principles to PepX, without accounting for sequence/structural redundancy. PepBind contains 3100 protein peptide complexes. While larger databases may help in assessing the applicability of docking methods on predicting longer peptides, the databases are not well curated (e. g. complexes in the databases might contain non-interacting chains, small molecule ligands or ions which might lead to erroneous assessment of docking tools (Wen *et al.*, 2018)). To overcome this limitation, a curated database *PepBDB* (Wen *et al.*, 2018) was developed, containing peptides up to 50 amino acids, with nearly 13,000 complexes in the dataset. A more recent database, *Propedia (*Martins *et al.*, 2021), contains over 20,000 high resolution complexes with peptides ranging from 2 to 50 residues. Propedia features a hybrid clustering based on sequence, interface structure

**Table 1.** Summary of the popular protein–peptide complexes datasets that are widely used for testing and benchmarking different docking tool

| Dataset | Number of complexes | Length of peptide | Special Features | Specific application | Availability |
|---------|---------------------|-------------------|------------------|---------------------|--------------|
| LEADS-PEP | 53 | 3–12 residues | Diverse sequence of peptides, complexes do not interact with nucleic acids | Due to smaller peptide size, suitable for testing tools adapted from small molecule docking tools | www.leads-x.org |
| PeptiDB | 105 | 5–15 residues | Diverse secondary structure of peptides including conformational change upon binding, complexes with diverse biological functions | Suitable for testing tools that tackle peptide flexibility | RCSB code of the complexes: https://ars.els-cdn.com/content/image/1-s2.0-S096921260900478X-mmc1.pdf |
| PPDbench | 133 | 9–15 residues | Diverse in term peptide sequences (<40% sequence similarity) and biological functionalities | Suitable for testing docking tools on different complexes categorised with different functionalities | https://webs.iiitd.edu.in/raghava/ppdbench/ |
| PepPro | 89 | 5–30 residues | Contains 58 unbound receptors structures | Useful for testing tools whether they can predict apo-holo conformational change | http://zoulab.dalton.missouri.edu/PepPro_benchmark |
| Propedia | ~20000 | 2–50 residues | Contains subsets of complexes based on clustering on different features such as sequence, interface structure or binding site | Broader range of peptide length allows it to test different type of docking tools. Also, different subset gives flexibility to user on testing their tools | https://bioinfo.dcc.ufmg.br/propedia |
| PixelDB | 1966 | NA | Uses machine learning to identify protein and peptide. This helps to overcome the issue of incorrectly identifying them when peptide is larger than the receptor | Broader range of peptide length allows any docking tools to be tested on | https://github.com/KeatingLab/PixelDB |

or binding site that retains a lower number of clusters (1,845, 1,891 or 1,466, respectively), allowing the user to be flexible for benchmarking purposes. Finally, *PixelDB* (Frappier *et al.*, 2018) contains close to 2,000 high resolution and non-redundant complexes. Unlike previous databases, this one relies on a machine learning algorithm along with a chain length cutoff to identify the receptor and peptide in a complex. This overcomes the issue of defining protein and peptide, for cases where the receptor size is smaller than its peptide binder.

## Sampling

Docking methods can be classified in two major categories depending on their use of templates for modelling the complex (template-based and template-free docking).

### *Template-based docking*

Template-based docking methods take advantage of known structures of either protein monomers or complexes and extract structural features to predict the unknown peptide–protein complex structure. We summarise these methods in Table 2 and provide further detail below. Template-based methods are grounded on the premise that the PPI interface is conserved and similar to either the

PPI interface or the different interacting fragments in a protein. Based on conserved interfaces, these methods build a modelling scaffold for the target systems. Indeed, 80% of the peptide–protein interfaces can be derived from fragment interactions in monomeric proteins (Obarska-Kosinska *et al.*, 2016). The first step of these class of methods is finding suitable templates for the target system from different databases. The most popular template-based docking method is *GalaxyPepDock* (H. Lee *et al.*, 2015). It takes a peptide sequence and a receptor structure as inputs to search for structural similarity in the PepBind database. It then uses a score ($S\_complex$) for each hit in the database, which is calculated combining the TM score of the receptor in the database with respect to the target receptor, and an interaction similarity score which is calculated based on the protein structure, peptide sequence and the interacting residue pairs (H. Lee *et al.*, 2015; 'Modeling Peptide–Protein Interactions, Methods and Protocols', 2017). The top 10 scoring templates with scores higher than 90% of the maximum score in the database are selected. These templates are then used to build models with GalaxyTBM predicting an estimated accuracy for each model. Testing on the PeptiDB database where it predicted 37 out of 57 complexes with acceptable or higher quality. In general, the GalaxyPepDock approach is quite reliably when the receptor TM score is greater than 0.7 (H. Lee and Seok, 2017). GalaxyPepDock takes 2–3 hours to complete a prediction, making it fast compared

**Table 2.** Summary of highlighted templated based docking tools

| Tool | Input | Link to Server/ Standalone | Peptide Flexibility | Receptor Flexibility | Specific applications/ Best cases to apply on |
|---|---|---|---|---|---|
| Galaxy PepDock | Protein structure +peptide sequence | Server: https://galaxy.seoklab.org/ cgi-bin/submit.cgi?type= PEPDOCK | Full flexibility at the refinement stage | Full flexibility at the refinement stage | Tested on the PepBind dataset. Predictions are reliable when templates can be found with TM score > 0.7 |
| PepComposer | Binding site information | Server: http://biocomputing.it/ pepcomposer/webserver | Sidechain rotamer and small change in backbone | Sidechain rotamer and small change in backbone | Suitable for small peptides, when tested on the LEADS-PEP dataset with 50% successes. Can also be used as inhibitor peptide design tool. |
| InterPep2-Refined | Protein structure +peptide sequence | Standalone: http://wallnerlab.org/ InterPep2 | SC flexibility at the refinement stage | Full flexibility at the refinement stage | Predictions are reliable when templates can be found with TM score > 0.7. Overall performs slightly better than GalaxyPepDock |

to methods that rely on computationally expensive molecular simulations. The recently developed InterPep2 (and InterPep2-Refined) also belongs to the template-based category, with a similar performance to GalaxyPepDock when tested on unbound receptors set and slightly better when tested on the bound dataset (Johansson-Åkhe *et al.*, 2020).

*PepComposer (*Obarska-Kosinska *et al.*, 2016) is an example of tool which uses structural knowledge of complexes to design a novel peptide sequence and dock it to the given receptor. This method, in the first step, finds a structurally similar fragment based on a given binding site and retrieves continuous backbone fragments from a structural database based on contacts to the prior fragment. In the next step, it predicts novel peptide sequences and bound complex structures using Monte Carlo moves embedded in a python-based tool (pyRosetta (Chaudhury *et al.*, 2010)). Testing the method on the LEADS-PEP dataset returned a 50% success rate considering only the top model. However, the caveat here is that the designed peptides are generally shorter than the native which decreases the RMSD (Root Mean Square Deviation) value (Obarska-Kosinska *et al.*, 2016).

### Template-Free Docking

This class of methods samples different peptide–protein orientations and positions as well as generates a diverse set of peptide conformations. Depending on the available knowledge of the binding site, these methods can be further divided into two subclasses: local and global docking. In local docking, the binding site is known, reducing search space. In global docking, no prior knowledge of the binding site is used, and the peptides explore the whole receptor surface (see Fig. 1). We will discuss these two subgroups in two different sections.

### Local Docking

As input, local docking methods need a set of user-defined information about the binding site, restricting the search of the ligand to the vicinity of this region, so we only need to sample the peptide conformations and orientations. Success of this group of docking

methods depends on the accuracy of the initial information (Ciemny *et al.*, 2018). Knowledge of the binding site can come from diverse sources such as protein–protein complex interfaces, a docked pose from another docking tool, hotspot prediction or even experimental data. Each method has its own specific input and types of information it can handle, requiring the right fit between prior information and local docking programme used. There are also limitations regarding how much sampling of the protein and peptide conformations are needed. As an example, *HADDOCK* uses Ambiguous Interaction Restraints based on hotspot residues (e.g. from NMR chemical shift perturbation data) on the protein surface to guide sampling – but requires different peptide conformations as input (e.g. alpha helix, extended and polyproline II) to limit sampling to the relative position of the protein/peptide without sampling the peptide conformations (Trellet *et al.*, 2013, 2014; Geng *et al.*, 2017). Once the preferred bound conformation is found, HADDOCK introduces peptide backbone flexibility to sample diverse conformations. HADDOCK has a 14.5% success rate when tested on the PeptiDB database. *HPEPDOCK*-local, using binding site hotspot information and shape complementarity followed by energy minimisation, produces fast and accurate predictions with 33.9% success rate when tested under the same condition as of HADDOCK (P. Zhou *et al.*, 2018; Johansson-Åkhe *et al.*, 2019). HPEPDOCK relies on *MODPEP* to generate an ensemble of unbound peptide conformations. An advantage of HADDOCK over HPEPDOCK is its ability to handle ambiguous data for the binding site (Williamson, 2013; Deplazes *et al.*, 2016). Finally, Rosetta FlexPepDock *ab initio* can also produce a diverse set of peptide conformations for binding, but this comes at a higher computational expense (Raveh *et al.*, 2010, 2011).

The second group of local docking methods are derived from small molecule docking programmes (such as *AutoDock Vina* (Rentzsch and Renard, 2015), *GOLD* (Verdonk *et al.*, 2003) or *Surflex-Dock (*Spitzer and Jain, 2012) to name a few). Although no initial conformation of the peptide is required, the accuracy rapidly decays when sampling beyond 10 flexible bonds, limiting peptide size (Ciemny *et al.*, 2018). In this approach, the peptide is placed in the binding site and peptide conformations are sampled

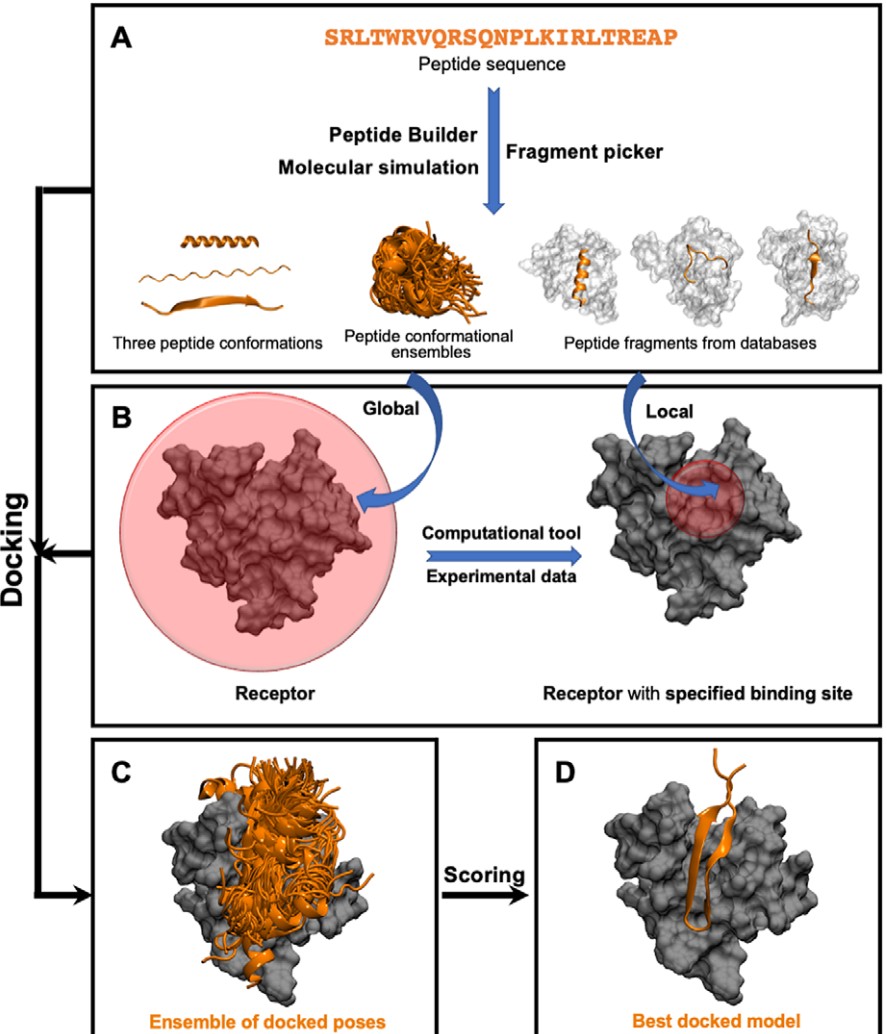

**Fig. 1. Pipeline in popular template-free docking methods.** (A) Input peptide conformations are generated in 3 major ways: 1) using peptide builder to generate major 3 conformations (alpha, polyproline II, extended); 2) molecular simulations are used to generate an ensemble of peptide conformations and 3) fragment pickers are used to select peptide fragments in the structural databases based on the peptide sequence. (B) If the binding site known, peptides are guided towards the binding site (local docking); else, peptides explore the whole protein surface (global docking). (C) Ensemble of docked poses. (D) Top score docked model representing the native structure.

using either Monte Carlo moves (AutoDock Vina and GOLD) or rotamer libraries (Surflex-Dock). AutoDock is most reliable with peptide lengths between 2 and 4 amino acids. Several methods have been developed to tackle longer peptides through incremental docking approaches (e.g. *DINC 2.0* (Antunes *et al.*, 2017), *DLPep-Dock* (Sun *et al.*, 2021)) (Antunes *et al.*, 2017). The incremental pipeline in DINC 2.0 has several stages: 1) dock a small fragment (preferably 6 rotatable bonds, roughly 2 amino acids) with Auto-Dock 4; 2) increase the peptide size by adding 3 more rotatable bonds and freezing 3 of the 6 previous rotatable bonds and 3) dock the peptide again (Antunes *et al.*, 2017). Using this approach DINC 2.0 has been successful with up to 25 flexible bonds. The selection of the initial fragment is done based on heuristics, while the extension of the fragment follows the potential to maximise H-bonding with the receptor. The benchmark test included a custom dataset of 73 protein peptide complexes with multiple successes, including the docking of a B2 chicken MHC class I receptor and an 8-mer chicken peptide (1.61 Å RMSD from the native structure). The *Glide SP-PEP* method also uses fragment-based docking with Itera-tive Residue Docking and Linking to dock peptides smaller than

8 amino acids (Diharce *et al.*, 2019). It uses Glide's SP-PEP module to dock each residue iteratively to the binding site and then uses the covalent module to create bonds between them. The success rate for this method was high in a custom-made benchmark set with 10 out of 11 successful docking examples.

The third group of local docking methods can be termed as refinement methods (*DynaDock* (*Antes, 2010*), *PepCrawler* (Donsky and Wolfson, 2011), *Rosetta FlexPepDock* (Raveh *et al.*, 2010)) or peptide inhibitor design methods (*PepCrawler*) rather than strictly docking tools. These methods need input structures of either coarse peptide–protein complexes (for refinement) or pro-tein–protein complexes (for inhibitor design) (Ciemny *et al.*, 2018). DynaDock, in the first step, generates broad sampling of the peptide conformation at the binding site by performing random rotation of backbone torsion and sidechain. In the next step, it uses an MD (OPMD) based refinement of the bound modes which allow the full flexibility to the receptor (Antes, 2010). Rosetta FlexPepDock uses Monte Carlo moves to sample diverse peptide conformations with full receptor flexibility with on-the-fly energy minimisation (Raveh *et al.*, 2010). Unlike these two methods, PepCrawler can be

used in two ways – refinement and inhibitor peptide design. For refinement, it uses an initial protein peptide complex structure as input and samples a diverse range of peptide conformations with a Rapidly exploring Random Trees (RRT) based algorithm. To design inhibitor peptides, it uses provided protein–protein complexes to generate the peptide fragment with lowest binding energy in the first step, followed by the RRT algorithm as refinement to dock the diverse peptide conformation. PepCrawler allows peptide and protein sidechains flexibility and only the peptide backbone flexibility (Donsky and Wolfson, 2011). This group of methods produce best results for short peptides (< 15 amino acids) and when the initial conformation of the peptide is below 5 Å RMSD from the native protein peptide complex (Ciemny *et al.*, 2018).

Some of the popular local docking tools are summarised in Table 3, together with suggested applications where each method is most successful. The first two groups of docking methods can be used when we only have binding site information and do not have any structural information of the complex. They are often used to generate initial models which can be later refined with other methods. Tools such as *PeptiMap* (Lavi *et al.*, 2013), *PepSite* (Trabuco *et al.*, 2012), *PEP-SiteFinder* (Saladin *et al.*, 2014), *SPRINT-str* (Taherzadeh *et al.*, 2017), *ANCHORSmap* (Ben-Shimon and Eisenstein, 2010) or *InterPep* (Johansson-Åkhe *et al.*, 2019) can predict the binding site, when no other structural information is available. The most recent method, InterPep uses template-based knowledge and a machine learning-based model to predict the binding site, outperforming most the other existing tools (Johansson-Åkhe *et al.*, 2019). Experiments such as Chemical Shift Perturbation, alanine scan mutagenesis or ligand foot printing mass spectrometry provide information about the binding site and can be used alternatively to binding site predictors.

## Global Docking

This class of template-free docking programmes becomes specially useful when there is no information about the binding site (see Fig.1). This class of methods is the most general as it requires the least amount of information provided by the user. However, the additional computational effort required to simultaneously sample the binding site as well as peptide conformations limits the success rate when compared to the previous classes of methods. Many of these approaches use a two-step procedure composed of a fast rigid docking stage to identify the bound state followed by a refinement strategy. Thus, the local docking methods described above can be used as part of the refining strategy.

There are several strategies to generate initial peptide conformations for the rigid docking stage: 1) methods such as *MDockPeP* (Yan *et al.*, 2016), *MDockPeP2* (Xu and Zou, 2022), *Cluspro PeptiDock* (Porter *et al.*, 2017), use a *MODELLER* (Webb and Sali, 2014) based algorithm and *PIPER-FlexPepDock* (*PFPD*) (Alam *et al.*, 2017) uses the Rosetta fragment picker to extract a fragment from an interacting partner of a protein–protein complex with similar sequence. MDockPep2 additionally considers the physiochemical environment similarity of the binding interface along with sequence similarity in the fragment picking stage; 2) *pepATTRACT* (Schindler *et al.*, 2015; Vries *et al.*, 2017) threads through three major secondary conformations (i.e. alpha, beta and coil) using *PeptideBuilder (*Tien *et al.*, 2013); and 3) *HPEPDOCK* global uses MODPEP to generate an ensemble of peptide conformations (P. Zhou *et al.*, 2018). Once the peptide conformation is identified, each method relies on their own rigid docking strategies. Cluspro PeptiDock and PFPD use a PIPER-based protocol (Kozakov *et al.*, 2006); MDockPeP uses a modified version of AutoDock vina

whereas MDockPeP2 uses ZDock (protein–protein docking tool) to carry out the rigid docking step (Xu and Zou, 2022; Yan *et al.*, 2016); pepATTRACT uses ATTRACT to carry out rigid body docking of the peptide with the ATTRAC coarse-grained representation of the protein and peptide and HPEPDOCK uses a modified version of MDock making it suitable for protein–peptide systems to perform rigid body docking. At the end of the rigid docking step, these methods have their own ways to include flexibility in the system and refine the docked structures. These strategies include using local docking methods, MD or MC simulations or other energy minimisers. For instance, PFPD uses Rosetta FlexPepDock; pepATTRACT uses iATTRACT and AMBER MD simulation for refinement and HPEPDOCK uses a SIMPLEX energy minimiser as the fully flexible refinement step (Schindler *et al.*, 2015; Alam *et al.*, 2017; P. Zhou *et al.*, 2018).

Methods like *CABS-dock (*Kurcinski *et al.*, 2015, 2019), *AnchorDock (*Ben-Shimon and Niv, 2015) and *AutoDock CrankPep* (ADCP) (Zhang and Sanner, 2019) allow flexibility to the peptide during the whole docking process. CABS dock generates peptide conformations in explicit solvent in the presence of the interacting partner, allowing the peptide to adopt its bound conformation. Thus, this allows full flexibility on both the receptor and peptide side. It uses a coarse-grained representation of each amino acid where backbone and sidechains are represented by two pseudo atoms each for computationally efficiency (Kurcinski *et al.*, 2015, 2019). The limitation in this case is the need for the peptide's secondary structure, which is not available in most cases. PSIPRED (McGuffin *et al.*, 2000) is used when the secondary structure is not known – even though it is not ideal for predicting the secondary structure of peptides, which are typically intrinsically disordered in their free form (Yan *et al.*, 2016). Unlike other global docking tools, AnchorDock uses the prediction tool *ANCHORSmap* to identify the anchoring spot and then performs an anchor-guided MD simulation (Ben-Shimon and Niv, 2015). This strategy combines the speed-up of the restraints with full flexibility of the peptide/protein system. A recent method, *ADCP*, uses Monte Carlo moves to sample peptide conformations under the influence of the potential landscape generated by the receptor which helps it to find correct fold upon binding making it highly successful allowing fully flexible docking (Zhang and Sanner, 2019).

Methods which use molecular simulations (extensive MD or MC) either in the docking stage or in refinement stage generally have higher accuracy than the other methods (Agrawal *et al.*, 2019; J. Wang *et al.*, 2019; Weng *et al.*, 2020). For example, despite the global nature of pepATTRACT, it can be as successful as some of the local docking approaches, with a local version of the method (pepATTRACT-local) having a higher success rate (Schindler *et al.*, 2015). PeptiDock+ Gaussian accelerated molecular dynamics (GaMD) which refines Cluspro PeptiDock results with Gaussian accelerated MD (discussed in the MD section), performs significantly better than the traditional Cluspro PeptiDock (J. Wang *et al.*, 2019). AnchorDock correctly predicted 10 of 13 complexes (RMSD <2.2 Å) in a custom dataset (Ben-Shimon and Niv, 2015). ADCP has shown better performance than most of the other existing docking tool for peptide ranging 16–20 residues with an 87% success rate (considering 10 models) when tested on LEADS-PEP dataset (Zhang and Sanner, 2019). Finally, *PFPD*, considered one of the state-of-the-art methods, can produce near native complex structures in 70% for bound test sets and 40% for unbound test sets (Alam *et al.*, 2017). However, due to the nature of the simulation, these methods are significantly slower compared to others. The computational resource requirements increase with longer

**Table 3.** Summary of highlighted 'local docking' tools. Here, acronyms are used as follows: Pstr, protein structure; pseq, peptide sequence; pconf, initial peptide conformation; BB, backbone; SC, sidechain

| Tool | Input | Link to server/standalone | Peptide flexibility | Receptor flexibility | Specific applications/best cases to apply on |
|---|---|---|---|---|---|
| HADDOCK | Pstr+pconf ensemble + ambiguous information of binding site | Server: https://wenmr.science.uu.nl/haddock2.4/submit/1 | SC are flexible but can be extended to the BB of the provided binding site residues | Fully flexible | HADDOCK can use ambiguous information about binding residues on protein and/or peptide. Reliable when there is no significant peptide conformational change upon binding. |
| HPEPDOCK -local | Pstr + pconf ensemble+ Information of binding site | Server: http://huanglab.phys.hust.edu.cn/hpepdock/ | Not flexible | Flexibility is considered generating an ensemble of peptide conformation | 34% success rate on PeptiDB database compared to HADDOCK's 14.5% but needs accurate information of binding residues |
| AutoDock Vina | Pstr+pseq+Binding site coordinate | Standalone: https://github.com/ccsb-scripps/AutoDock-Vina | SC flexibility is default but can be extended to the BB | Fully flexible | Reliable when binding peptide length is less than 5 residues |
| DINC 2.0 | Pstr + pconf + Binding site coordinate | Server: http://dinc.kavrakilab.org | No flexibility | Fully flexible | AutoDock based method with fragmentation of peptide. This allows it to tackle peptides up to 8 residues |
| PepCrawler | Initial coarse protein–peptide with peptide at the binding site/protein–protein complex | Server: http://bioinfo3d.cs.tau.ac.il/PepCrawler/php.php | SC flexibility | Fully flexible | Can be used as a refinement method. Predictions are reliable when the starting model is with 5 Å RMSD from the experimental structure, and peptide is shorter than 15 residues |
| Rosetta FlexPep Dock | Initial coarse protein peptide complex with peptide at the binding site | Server: https://www.sciencedirect.com/science/article/pii/S1359644617305937#bib0165 | SC flexibility but can be extended to the BB | Full flexible | Can be used as a refinement method. Predictions are reliable when the starting model is with 5 Å RMSD from the experimental structure, and peptide is shorted than 15 residues |

peptides while maintaining the accuracy (e.g. simulations running for hours on GPUs or longer in CPUs) (Ciemny *et al.*, 2018). These expensive simulations are needed to achieve the higher accuracy as shown by the pepATTRACT web server which removes the refinement step (Vries *et al.*, 2017). When using the PeptiDB benchmark set, the webserver predicts 14 out of 80 complexes correctly (within 2 Å RMSD of the experimental structure), whereas the full pepATTRACT version correctly predicts 38 complexes (Vries *et al.*, 2017).

Recently, the Furman lab introduced *patchMAN*, a motif search method (using the MASTER algorithm) combined with Rosetta FlexPepDock refinement, which outperforms other methods (Khramushin *et al.*, 2022). At the first step, it searches for the receptor surface motifs in a non-redundant protein database followed the finding peptide templates that interact with these motifs and the target peptide sequence is threaded through the models. In the end, Rosetta FlexPepDock is used to refine all the models. This method allows fully flexibility of the binding site of the receptor as well as the peptide. When tested on the custom-made PFPD database, it outperforms PFPD and even recent machine learning-based revolutionary AlphaFold (Jumper *et al.*, 2021), considering the success criteria as a 2 Å RMSD cutoff from the native. However, on a different dataset (LNR) its performance is comparable to AlphaFold (Khramushin *et al.*, 2022). Table 4 summarises the global docking tools listing their features and suitable application cases.

## Scoring

At the sampling step, docking methods obtain an ensemble of docked poses – some of them are native-like, while some are far from native. The state of the art in peptide docking is reliable at sampling the correct binding site. For example, when we only consider sampling efficiency, MDockPeP has success rate of 95% when starting from bound conformations and 93% when starting with challenging unbound structures (Yan *et al.*, 2016). The recent method patchMAN can sample within 5 Å RMSD from the native complex in 100% cases (Khramushin *et al.*, 2022). This implies that currently, the limitation and overall successes of the docking tools can be attributed to the scoring stage majorly. Thus, the next crucial step is to find the best docked model, representing the native complex, in the ensembles of the docked poses (Ciemny *et al.*, 2018; Weng *et al.*, 2019) (see Fig. 1). These methods can be classified in several major groups such as using a knowledge-based scoring function, energy-based method, clustering-based method and integrative or combinational approach. One important feature of a good scoring method is that it should consider entropic contribution due to conformational change as well as the interaction energy.

There are a series of scoring function used successfully in small molecule and protein–protein docking field. End point methods like MM/PBSA and MM/GBSA, mentioned in the MD section, are widely used for small molecule binding-free affinity calculation and scoring (Hou *et al.*, 2011; Pu *et al.*, 2017; E. Wang *et al.*, 2019; Weng *et al.*, 2019). When applied to peptide–protein systems, if these methods are used with appropriate parameter, they outperform pepATTRACT (which uses ATTRACT scoring function) and produce similar quality as HPEPDOCK-local (which uses an iterative knowledge-based scoring function coming from protein–protein docking tool MDock) (Weng *et al.*, 2019). However, these methods do not consider entropic contributions due to peptide conformational changes, limiting their success to binding processes without significant changes in the peptide conformation. Ideally,

these methods should be modified or combined with others for generalised use (Spiliotopoulos *et al.*, 2016; Tao *et al.*, 2020). As an improvement, BiPPred and HADDOCK use a dampened version of MMPBSA named dMMPBSA algorithm as a scoring function to calculate the free energy and rank docked poses (Spiliotopoulos *et al.*, 2016). In this approach, they reduce the Coulombic interaction and polar solvation term by factor of 5 to compensate the overestimation of free energy due to the omission of entropy. Another recent approach by H. Tao *et al.* combines MM-GBSA scoring function with knowledge-based scoring function ITScorePP to consider the conformational entropy part (Tao *et al.*, 2020). ITScorePP is derived from atomic distance-based energies parametrised iteratively using statistical mechanics. Their work has shown rescoring the pose clustering with this combined scoring function makes the results significantly better when compared against pepATTRACT, CABS-dock and HPEPDOCK result with LEADS-PEP dataset.

A group of methods use clustering algorithms on the ensemble of docked poses (or filtered docked poses) based on structural RMSD – but they have their own way to use clustering for selecting structure. Cluspro PeptiDock assigns highest score to the most populated cluster's medoid, whereas CABS-dock selects the consensus medoid obtained from different clustering protocols as the best model (Kurcinski *et al.*, 2015, 2019; Porter *et al.*, 2017). Rosetta FlexPepDock and PFPD perform clustering and score the top clusters with a modified Rosetta ab-initio energy function (Alam *et al.*, 2017; Raveh *et al.*, 2010). The modified version of the Rosetta ab-initio energy function has been shown be successful as it combines standard all atom Rosetta energy with internal peptide energy and interaction energy. AnchorDock also uses clustering algorithm on all snapshots from molecular simulation trajectories and scores the clusters based on the average potential energies of the best 15 models (in terms of binding energy) in each cluster (Ben-Shimon and Niv, 2015). pepATTRACT's ATTRACT scoring function is based on modified Lenard–Jones function to select 1000 models and those are further refined by AMBER followed by performing clustering on simulated trajectory. The clusters are ranked based on the average ATTRACT energy of four lowest energy models (Schindler *et al.*, 2015).

Integrative scoring methods combine external information like agreement with co-evolutionary data or mutagenesis data with energy-based or clustering-based scoring which have performed very well in the recent CAPRI competitions (Yu *et al.*, 2017; Lensink *et al.*, 2020). Machine learning is recently becoming one of the methods of choice to derive scoring functions. InterPepRank (Johansson-Åkhe *et al.*, 2021) is such an example which uses deep graph-based neural network mapping the protein peptide complex, and DockQ (Johansson-Åkhe *et al.*, 2020) uses a Random Forest model to score and rank the docked poses. Recently, the field of docking has started using combined multiple scoring method together to compensate each other limitation. For example, when InterPepRank is combined with PFPD pipeline as rescoring, success rate increases significantly (40% for the high-quality prediction). Simultaneously, it filters out some of the non-native dock poses using an InterPepRank score cutoff, reducing the number of hits for refinement steps which increases the computation efficiency (Johansson-Åkhe *et al.*, 2021).

## Summary

The wealth and diversity of available docking programmes and servers for peptide–protein complexes pose a barrier of entry to

**Table 4.** Summary of highlighted 'global docking' tools. Here, acronyms are used as follows: Pstr, protein structure; pseq, peptide sequence; BB, backbone; SC, sidechain

| Tool | Input | Link to Server/standalone | Peptide Flexibility | Receptor Flexibility | Specific applications/ Best cases to apply on |
|---|---|---|---|---|---|
| MDockPeP | Pstr+pseq | Server: https://zougrouptoolkit.missouri.edu/mdockpep/ | Small change in conformation at the refinement stage | Full flexibility at the refinement stage | Performs well on smaller peptides with <15 residues |
| MDockPeP2 | Pstr+pseq | Standalone: https://zougrouptoolkit.missouri.edu/mdockpep2/download.html | Full flexible | Fully flexible at the refinement stage | Can be applied on peptides up to 29 residues but success rate decreases beyond 15 residues |
| Anchor Dock | Pstr+pconf | Not available | Fully flexible | Fully flexible | Uses expensive molecular simulations. Suitable for large peptides (>15 residues) which show conformational changes |
| Pep ATTRACT | Pstr+pseq | Server: https://bioserv.rpbs.univ-paris-diderot.fr/services/pepATTRACT/ | Fully flexible in the full pepATTRACT version but no flexibility in the web server | Fully flexible in the full pepATTRACT version but the server just uses 3 major peptide conformations to dock | Full version uses expensive molecular simulations. Suitable for large peptides (>15 residues) which shows conformational changes. Web version is useful for smaller peptides |
| CABS-dock | Pstr+pseq+Bound peptide secondary structure (optional) | Server: http://biocomp.chem.uw.edu.pl/CABSdock | Fully flexibly at the peptide conformation generation stage | Fully flexible | Suitable when bound peptide conformation is known |
| PIPER-FlexPep Dock | Pstr+pseq | Server: http://piperfpd.furmanlab.cs.huji.ac.il | Fully flexible at the refinement stage | Fully flexible at the refinement stage | Uses expensive molecular simulations. Suitable for large peptides (>15 residues) which shows conformational changes |
| AutoDock CrankPep | Pstr+pseq | Standalone: https://github.com/ccsb-scripps/ADCP | Fully flexible | Fully flexible | Uses expensive molecular simulations. Suitable for large peptides (>15 residues) which shows conformational changes |
| patchMAN | Pstr+pseq | Server: https://furmanlab.cs.huji.ac.il/patchman/ | Fully flexible at the refinement stage | Fully flexible at the refinement stage | Most successful when tested on custom made PFPD dataset outperforming AlphaFold |

newcomers to the field or would-be users. It is difficult to answer a question like, 'which is the best?' as it will depend on the system and available information. Each method has advantages and disadvantages such as the ability to work on peptides of different size, or the ability to explore large conformational changes upon binding. In general, local docking tools outperform the global tools, but the latter does not need any information about the binding site. Most of the global docking tools can be used as local docking tools when the binding site information is provided leading to higher success rates

(Schindler *et al.*, 2015; P. Zhou *et al.*, 2018). A benchmark study by Weng et al. showed that performance of local docking methods that are based on AutoDock drops significantly depending on erroneous inputs of the size of binding sites, especially for peptides longer than 10 residues (Weng *et al.*, 2020). In summary, for a user when the target receptor can find a template with enough homology (TM score > 0.7), template-based methods generally offer the highest success rates. Alternatively, when the TM score is lower than 0.7, template-free methods should be used. For peptide smaller

than 5 residues, the docking tools coming from small molecule docking work the best (Ciemny *et al.*, 2018). For the longer peptides, computationally extensive molecular simulation-based tools such as PFPD, or ADCP and patchMAN become more relevant. Running short MD simulation-based refinements have shown higher successes; however, computational expenses also rise. In future, coarse graining on the non-interacting residues might be a way to reduce the computer time required for these molecular simulation-based tools. Moreover, the docking field has started to apply combined approaches like combined predictions from Inter-Pep2-refined and PFPD which has shown better performance than the individual one (Johansson-Åkhe *et al.*, 2020).

## MD section

The nature of MD simulations allows, in principle, to obtain thermodynamics, kinetics and mechanistic understanding of the peptide–protein binding and unbinding process. As in the application to other biological problems, MD is limited by the accuracy of the physics model used and the ability to sample the complex energy landscape, which typically requires computational resources beyond our current capacity.

## Modelling PPIs with modern force fields

The interactions of molecules can be theoretically determined using quantum mechanics but remain unaffordable in practical terms for large biomolecules. In practice, an empirical force field is used to model such interactions along with Newton's equations of motion to simulate the dynamics. MD simulations have been shown to accurately predict the binding potency of diverse small molecule binders (L. Wang *et al.*, 2015). However, there are key differences between protein binding with small molecule and peptide that makes the latter more computationally challenging. First, while small molecules have a few hotspot interactions that dominate recognition, peptide recognition can stem from many weak interactions. Second, the structure of peptides can be highly flexible, requiring finely tuned parameters that are sensitive to the change in conformational preferences between the free conformation of peptide and its bound form. Thus, peptides have been specially affected by known biases in secondary structure preferences present in some force fields (Perez *et al.*, 2015; Robinson *et al.*, 2016).

Early force field development was an art, guided by great scientific insights (e.g. some parameters originated from 'guesses' that have remained as part of the force field for decades), and carried out by a few expert groups. One of the challenges is the unexpected consequence of parameter changes as modifying one parameter might affect the accuracy of another parameter that was not adjusted due to the coupling between different terms. Even long MD simulations on a set of systems with times series, distributions, behaviour and stability analysis might not be enough to capture all possible issues. Some issues might arise in timescales beyond those studied during development or in systems not included in the benchmark test. While some groups might have made *ad hoc* modifications to deal with problems in specific systems, these modifications were not often properly benchmarked and rarely made it back into the main force field branch. Such trend has dramatically changed in recent years as measured by the number of force fields as well as involvement of many groups representing these improvements. Despite this, a general problem is the lack of a golden standard benchmark set for parameter development (e.g. for

proteins, nucleic acids, lipids). The availability of open sharing resources would make the preparation and dissemination of such a benchmark test an easy endeavour. Community efforts such as the OpenFF are already on their way for continuous optimisation of small molecule force fields (Qiu *et al.*, 2021).

Here, we focus on recent efforts to improve the description of peptides and intrinsically disordered proteins (IDPs) while maintaining the stable properties for folded proteins (Fig. 2), more extensive surveys can be found elsewhere (Rahman *et al.*, 2020; Mu *et al.*, 2021). Force field development follows three main strategies: 1) modifying the dihedral angle parameters against experiments and/or quantum mechanical calculation; 2) adjusting non-bonded protein–water interactions and 3) balancing dihedrals with grid-based energy correction maps (CMAP).

Most effort is concentrated on directly refining the global backbone dihedral parameters including AMBER ff99SB* and ff03* (Best and Hummer, 2009), ff14SB (Maier *et al.*, 2015), ff99SB-ILDN (Lindorff-Larsen *et al.*, 2010), CHARMM22* (Piana *et al.*, 2011), OPLS-AA/M (Robertson *et al.*, 2015) and OPLS3 (Harder *et al.*, 2016), while further improvements also involve refined side-chain dihedral terms. The residue specific force field (RSFF) approach involves residue-specific dihedral parameter refitting to achieve better agreement with experimental data. RSFF1 force field (Jiang *et al.*, 2014) is developed based on OPLS/AA (Kaminski *et al.*, 2001), while RSFF2 (C.-Y. Zhou *et al.*, 2015) is based on the ff99SB force field (Hornak *et al.*, 2006). Protein–water interactions are actively involved in peptide–protein binding process, where the ensemble of peptide conformations in its free form is highly sensitive to the solvation model used (e.g., overall compactness as defined by the radius of gyration). Not surprisingly, the AMBER ff03ws combined with a refined TIP4P/2005 solvation model (Best *et al.*, 2014) and the CHARMM36m with an optimised TIP3P model (Huang *et al.*, 2017) have shown improvement in modelling IDPs ensembles. While some combinations of solvent force fields are designed to work with specific protein force fields, others such as the TIP4P-D (Piana *et al.*, 2015) aim to improve general deficiencies such as underestimation of London interactions by developing larger water dispersion coefficient, resulting in improved agreement with experimental observables over a broad range of force fields. The CMAP strategy was first introduced as a grid-based correction to the CHARMM22 force field (MacKerell *et al.*, 1998) to account for coupling of ψ/φ torsion angles (CHARMM22/CMAP) (Mackerell *et al.*, 2004). The latest iteration (CHARMM36) (Huang and MacKerell, 2013) has been the starting point for new CMAP potentials that better balance between folded proteins and IDPs (CHARMM36m). The CMAP approximation has been adopted in other families of force fields such as AMBER (Perez *et al.*, 2015; Tian *et al.*, 2019) or OPLS (Yang *et al.*, 2019). Over recent years, the CMAP strategy has led amino acid specific potentials, rather than using a few transferable potentials for all (e.g., non-glycine or proline). Thus, ff99IDPs (W. Wang *et al.*, 2014) and ff14IDPs (Song *et al.*, 2017) develop specific CMAP parameters for eight disordered-promoting amino acids. More recently, ff14IDPSFF (Song *et al.*, 2017) and CHARMM36IDPSFF (Liu *et al.*, 2018) add a different CMAP correction to each of the 20 amino acids.

Although such optimisations can better describe the more extended conformation of disordered peptides, some modified force fields generate unstable structures for folded proteins. Ideally, a force field that allows accurate descriptions of both folded and unfolded ensembles is preferable because it would better simulate transitions of peptides between disordered state to ordered state. The a99SB-disp force field developed by Robustelli *et al.* modified

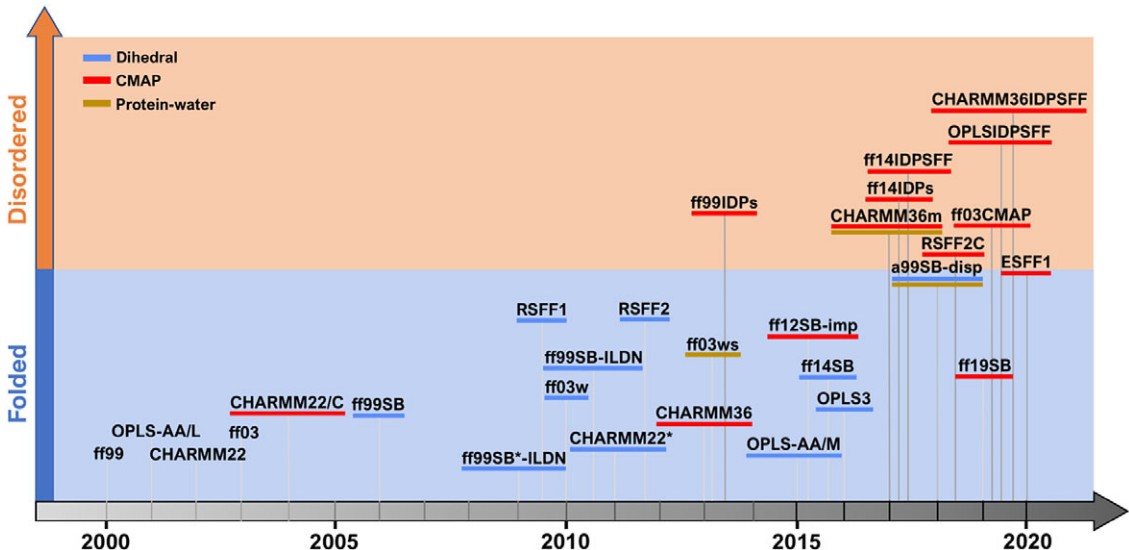

**Fig. 2. Overview of protein force field development after 2000.** Each protein force field is classified by the year of publication, target systems for optimisation (folded, disordered or both) and additional underscores indicating whether it is a modification version of previous force fields using strategies including dihedral parameter adjustment (blue), CMAP correction (red) or parameter modification for protein–water interaction (gold).

the ff99SB-ILDN parameters and adjusted TIP4P-D water model against experimental measurements, and the resulting force field has shown great improvement for modelling disordered ensembles and still maintains the accuracy for folded proteins (Robustelli *et al.*, 2018). Another environment specific force field (ESFF1) was recently developed based on CMAP corrections of 71 different sequence environments (Song *et al.*, 2020). These force fields have demonstrated an improved balance between modelling IDPs and folded proteins. With the number of choices available, it might be daunting to choose the right force field for your system. Many MD packages such as AMBER (Case *et al.*, 2005), CHARMM (Brooks *et al.*, 2009), NAMD (Phillips *et al.*, 2005), Gromacs (Abraham *et al.*, 2015), Tinker (Lagardère *et al.*, 2017) or OpenMM (Eastman *et al.*, 2017) provide user a wide selection of force fields, even originating from different force field families (e.g., AMBER or CHARMM family of force fields). It is important that for whichever force field is selected the compatible solvent model and ion parameters tested by benchmark studies should also be used correspondingly.

## Characterising peptide binding poses and affinities by MD simulations

MD is not high-throughput enough in its own to routinely determine the structures of peptide–protein complexes when the bound state is unknown. The three major applications for MD are:

1) Refinement of docking results; 2) estimating binding affinities based on known bound complexes and 3) use of integrative modelling strategies to determine structures of the complexes.

Docking approaches described in the previous section favour speed at the expense of accuracy, while MD approaches are accurate, but inefficient at identifying where and how a peptide binds from conventional MD simulations. Thus, short MD simulations are often the last step of docking calculations to eliminate steric overlap, account for local conformational changes and identify structures based on physico-chemical principles rather than relying on a scoring function. While this application is standard, it does not leverage the full potential of MD such as calculating binding

affinities. Thus, recent integration of docking and MD-based techniques such as the combination of ClusPro PeptiDock with GaMD goes beyond refining the structure to provide free energy profiles (J. Wang *et al.*, 2019). In this work, the authors benchmarked their method on three distinct model peptides achieving 0.6–2.7 Å improvement in peptide backbone structures. Moreover, the unbiased free energy profiles help identify key residues involved in significant conformational changes upon binding that can later be used for peptide sequence optimisation and design.

When the experimental structure of the complex is known, end-point methods based on MD are typically used to determine binding affinities. MM/GBSA and MM/PBSA are among the most popular methods in this category, introduced by the Kollman group two decades ago, the method is grounded on robust physico-chemical principles (E. Wang *et al.*, 2019). The method has been well received by the community and still favoured over empirical and semi-empirical scoring functions designed for protein–peptide docking (see discussion in docking section). Despite its robust theoretical framework, its practical implementation results in approximations (such as flexibility, solvation and entropy) that limit the accuracy of the results. For example, prediction of peptide binding affinities for peptide–MHC complexes is highly desirable for vaccine design, but the flexible nature of the peptides make routine affinity prediction using bioinformatic pipelines insufficient. The inclusion of structural information is crucial to explore relevant molecular conformations of the peptide–protein complex, and therefore, key to understand its dynamic behaviour. Wan *et al.* combined the MM/PBSA and the conformational entropy method to compute peptide–MHC binding affinities from MD simulations where both the bound and unbound peptide were simulated (Wan *et al.*, 2015). The method achieves highly correlated binding affinity rankings with experimental estimates after normalising $\Delta G_{MM/PBSA}$ with the hydrophobicity of peptides. Ochoa *et al.* generated conformations of the complexes from MD simulation, then using a scoring function to predict binding affinities in better agreement with experiments than either sequence-based predictions or single docking scoring methods (Ochoa *et al.*, 2019). Pathway-based free energy calculation methods such as free energy perturbation (FEP)

have achieved unprecedented accuracy in modelling protein binding with small molecule for a large set of ligands (L. Wang *et al.*, 2015). However, directly transferring such approaches to estimate protein–peptide binding free energy is challenging due to the flexibility and size of peptides. Kilburg *et al.* introduced a single-decoupling alchemical method that successfully calculated the free energy for HIV1-IN binding with a series of cyclic peptides (Kilburg and Gallicchio, 2018). The calculation convergence is largely affected by the ladder parameters in Hamiltonian and temperature replica exchange; specifically, more dense parameters are required to increase the overlap between phase space of alchemical states in large ligand system. FEP-based simulations have also been applied to estimate the mutation effect in binding specificity change of PDZ-peptide system (Panel *et al.*, 2018) and help select a potent blocker for Kv1.3 channel (Rashid *et al.*, 2013).

Integrative approaches combine computational modelling with experimental information to determine structures of peptide–protein complexes. Methods such as Rosetta and MODELLER are examples involving different types of modelling strategies. Other methods such as maximum entropy (Pitera and Chodera, 2012) or Bayesian inference (MacCallum *et al.*, 2015) aim to identify distributions that agree with experimental data. Analysing such distributions yields the number of states that best represent the data. Our use of the modelling employing limited data approach for peptide binding has been successful for harnessing chemical shift perturbation NMR (Mondal *et al.*, 2022)and ALA scan mutagenesis data in predicting conformations of the bound complex (Morrone *et al.*, 2017b). Furthermore, its physico-chemical foundation allows the user to recover relative binding-free energies using a competitive binding protocol. These simulations sample peptides conformations in binding while allowing their full flexibility and accurately match experimental results for a series of peptides inhibiting the p53-MDM2 and MDMX interaction (Morrone *et al.*, 2017a,b).

## Unveiling peptide binding/unbinding kinetics through enhanced sampling

Accurate prediction of peptide–protein binding/unbinding kinetics from MD simulations requires extensive sampling of bound/unbound states, the transitions between them and possible intermediate states. However, the structural flexibility of many peptides challenges estimation of association and dissociation rates, represented by $k_{on}$ and $k_{off}$, respectively, as complicated binding mechanism arises including folding upon binding of peptides to the receptor, and inherent structural heterogeneity arises from weak interactions in the binding interface. Recent simulation studies on PPIs employ various advanced sampling methods with critical thermodynamic and kinetic analysis. Markov state models (MSMs) have been widely applied to estimate kinetic quantities of biomolecular conformational dynamics from a set of short atomistic MD simulations (Chodera and Noé, 2014). Paul *et al.* used multi-ensemble Markov models, which combine conventional MD with Hamiltonian replica exchange enhanced sampling simulations, to characterise peptide–protein binding mechanism and kinetics beyond the seconds timescale of a nano-molar peptide inhibitor PMI to the MDM2 receptor (Paul *et al.*, 2017). Zhou *et al.* studied the p53 binding with MDM2 by running near 1 ms unbiased simulations on a distributed computing platform (G. Zhou *et al.*, 2017). Two key intermediate states were identified from a four-state kinetic model using MSM analysis and $k_{on}$ was predicted in good agreement with experimental estimation. Zwier *et al.* generated

hundreds of continuous binding pathways from weighted ensemble simulations and obtained similar on-rate estimates (Zwier *et al.*, 2016). In addition, they identified residue F19 from p53 might be a kinetically important residue for binding as the majority of conformations involve its partial or complete burial.

Metadynamics employs biasing potential as a function of collective variables by which the system is allowed to cross high-energy barriers that are conventionally difficult to sample (Bussi and Laio, 2020). Zou *et al.* investigated the folding and binding process of p53 to MDM2 using two metadynamics-based methods yielding a reasonable estimation for the on/off-rate constants and the binding-free energy profile (Zou *et al.*, 2020). The anchor residues F19 and W23 of p53 were identified to follow the stepwise binding pattern. This finding helps explain certain mutants can be regulated by weak non-native interactions near bound state due to the disorder nature of p53. The consequence of secondary interactions on the binding mechanism was also addressed by extensive unbiased simulations combined with umbrella sampling to perform MSM analysis for a coronavirus-derived peptide, bound to a prevalent MHC receptor in humans (Abella *et al.*, 2020). The model reaffirms the major role of anchor positions in the peptide for establishing stable interactions and reveals the underestimated importance of a non-anchor position. The conclusion was confirmed by simulating the impact of specific peptide mutations and validated these predictions through competitive binding assays where stark differences in unbinding pathways were identified by comparing the MSM of the wild-type system with those of the D4A and D4P mutants.

## Machine Learning

The role of machine learning in structural biology was greatly accelerated by the success of AlphaFold (AF) in the 13[th] installment of the Critical Assessment for Structure Prediction (CASP) event (Senior *et al.*, 2020). Two years later, after the field had replicated all the previous successes, a complete re-design of AF produced even higher accuracy structure predictions that surpassed any previous expectation (Baek *et al.*, 2021; Jumper *et al.*, 2021). The accuracy of such ML predictions is sometimes in better agreement with NMR data than the models generated by standard NMR pipelines (Tejero *et al.*, 2022). Not surprisingly, the field was soon ready to test the limits and possibilities of the AF approach. Early on, adding poly-glycine linker successfully tricked AF into predicting the structures of complexes where the linker remained unstructured. This strategy has produced a level of accuracy for peptide–protein complex structure prediction that surpasses state-of-the-art docking programmes in a recent benchmark test, especially for complexes with binding motifs. A retraining of AF for complexes was soon published online (AF-multimer), but the weights have not been as extensively refined as the ones for original AF (Evans *et al.*, 2021).

The success of such approaches begs the question of why AF is performing well on complexes such as peptides. One observation points to the structural complementarity between peptides and proteins, where many peptides adopt well-defined secondary structures upon binding. Indeed, peptides that bind as coils are not predicted as well, although generally the binding site is still identified. The field of docking uses scoring functions to rank the different poses and compounds – a strategy that is very successful for small molecules, but which has not reached the same level of maturity for peptides (as described above in the docking section).

AF's pLDDT measure also lacks the possibility of ranking different peptides as different sets of peptides might be predicted with similar pLDDT scores despite very different binding affinities. Other ML approaches directly use structural ensembles and a measure of accuracy such as RMSD to assess the quality of the predicted structures (Townshend *et al.*, 2021). This begs the question of whether AF or other ML algorithms can learn something about the biophysical energy function that governs binding (or folding) and how it can be used towards predicting peptide–protein complex structures.

AlQuraishi's group first addressed this question using a bespoke Hierarchical Statistical Modelling ML approach to learn the biophysical function that scores multiple peptides binding a receptor motif (Cunningham *et al.*, 2020). Unfortunately, lack of data meant that this approach could only be used for eight protein families. Could AF capture such as biophysical function from its training? Recent work from Ovchinnikov's group suggests that indeed AF has learnt such a function (Roney and Ovchinnikov, 2022). According to this study, MSAs serve the purpose of global sampling, focusing the search space in regions of interest, and the biophysical energy learned through the network is able to identify the best local structure. This is especially interesting for peptide–protein systems where the problem can be separated into two parts: (1) a template or MSA for the receptor and (2) a single sequence for the peptide. In this way, the learned function is responsible for finding where the peptide interacts, its conformation and any conformational changes required for the receptor. We put this notion to the test by using competitive binding in AF to determine which peptides had higher affinity to a series of receptors. Surprisingly, the method was very successful in ranking the strongest binders. As the differences in binding affinity become small, the method reflects this uncertainty, and the method is not suitable when both peptides are weak binders. A caveat of the above developments is the requirement to fit within the original hypothesis of peptide–protein complementarity – when the structure of the complex is not correctly predicted, then the competitive binding will also not work.

Turning this argument on its head, the Baker group uses the idealised version of proteins that neural networks learn to design new proteins. The process of deep neural network hallucination has already produced several structures that have been confirmed experimentally (Anishchenko *et al.*, 2021). Adding constraints into the hallucination process can direct the design into areas of complementarity or desired functionality. As such, the complementary nature of peptides to receptor binding motifs can lead to the design of peptides or mini-proteins based on constrained hallucination to the known binding site (J. Wang *et al.*, 2021).

The advances in ML for structural prediction and accurate scoring give rise to the ability to query increasing large libraries of peptides (Chang and Perez, 2022). Along these lines, PPI predictors are also starting to emerge to predict which peptides will interact with a certain protein and give insight into the peptide residues involved in the interaction (Casadio *et al.*, 2022; Lei *et al.*, 2021). Furthermore, ML is also offering ways to identify peptide sequences which are likely to have high biological activity against a particular pathology (Wu *et al.*, 2019) (e.g. anticancer (Chen *et al.*, 2021) or antimicrobial (Dee, 2022; E. Y. Lee *et al.*, 2017; Plisson *et al.*, 2020) peptides). Thus, we expect combining ML pipelines that act at the sequence level with those at the structural level will be able to create peptide libraries specific to a type of disease that can then be screened to predict structures and their relative binding affinities. This is a rapidly changing field, where ML is already having a big impact, and where many questions regarding the interpretability and applicability of the current technology need to be answered.

## Conclusions

Our focus in this review has been to identify the different approaches that docking, molecular simulations and machine learning use to study peptide–protein systems. We expect new synergies between these three types of technologies will lead to more robust methodologies to capture peptide–protein systems. For example, docking can reliably identify the binding region and might provide good templates for ML to refine, to predict structures and screen peptides for binding-free energies. Meanwhile, the emphasis of new force fields in correctly describing intrinsically disordered peptides together with enhanced sampling can benefit from initial models to determine kinetic constants and binding affinities through an orthogonal approach (on a more limited set of systems to refine). A current limitation in machine learning is the dependence on natural amino acids. MD on the other hand has transferable potentials and can be used as an end stage in peptide optimisation for studying the effect of controlling flexibility (e.g., through chemical staples) or replacing some residues with non-natural amino acids such as peptoids.

**Acknowledgement.** The authors thank XSEDE resources.

**Author contributions.** Conceived the review: AP.; Performed research and analysed data: AM, LC.; Wrote manuscript: A.M.,L.C. and AP.

**Conflict of interest.** The authors declare no conflicts of interest.

**Open Peer Review.** To view the open peer review materials for this article, please visit http://doi.org/10.1017/qrd.2022.14.

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
