## [Reviewer Report]

*Comments to Author*: “Modeling peptide-protein complexes: docking, simulations, and machine learning”

This review explores three main approaches in modeling peptide-protein interactions; molecular docking, molecular dynamic simulations, and machine learning. Overall it is a comprehensive and well written review that provides prospective users with the advantages and limitations of each approach. This kind of reviews is indeed of a great importance to the filed of computational chemistry as it lowers the barrier of entry to newcomers and guide them through choosing the correct approach. I suggest considering the publication of this review after performing the following minor revisions:

1. Molecular docking section: This section is well written and the authors provided a very good literature review of protein peptide-docking tools. My sole concern is that with the large amount of information provided here, the reader may find a difficulty in extracting information easily. Adding summary tables of each section can help the reader extract information efficiently. For example, the authors can provide a table summarizing the main features of each databases and sampling method [or at least the top performing], its pros, cons, and suggested application. Based on the scientific question, the prospective user can pick a database or sampling method that matches their system characteristics and refer to the the details in the main text.

2. Molecular dynamics section:

The authors provided a summary of the evolution and limitation of the current forcefields and explained the applications of different MD simulations methods in studying protein-peptide interactions. Although the authors explained the applications of several end-point free energy methods in protein-ligand/protein- peptide interactions such as MM-PBSA and MM-GBSA,they did not mention examples of using pathway free energy methods especially the highly rigorous “from statistical mechanics point of view” alchemical free energy approaches. Alchemical free energy calculations is a pathway free energy method that compute the binding free energy difference between the bound and unbound states of the peptide-protein complex by linking them through a suitable thermodynamic non physical “alchemical” path. Although the application of this approach in protein-peptide interaction is less common and more challenging than other end-point methods, it is worth shedding the light on the few successful examples of this approach as it will make this review more comprehensive [please refer to the following examples]:

1- Kilburg, D., & Gallicchio, E. (2018). Assessment of a single decoupling alchemical approach for the calculation of the absolute binding free energies of protein-peptide complexes. Frontiers in molecular biosciences, 5, 22.

2- Rashid, M. H., Heinzelmann, G., Huq, R., Tajhya, R. B., Chang, S. C., Chhabra, S.,& Kuyucak, S. (2013). A potent and selective peptide blocker of the Kv1. 3 channel: prediction from free-energy simulations and experimental confirmation.Â PloS one,Â 8(11), e78712.

3- Panel, N., Villa, F., Fuentes, E. J., & Simonson, T. (2018). Accurate PDZ/peptide binding specificity with additive and polarizable free energy simulations.Â Biophysical journal,Â 114(5), 1091-1102.

3. Machine Learning: No revision suggested.

---

## [Reviewer Report]

*Comments to Author*: Brief Summary:

Firstly, I would like to thank the Editorial Board of Quarterly Reviews of Biophysics, Discovery, for the opportunity to review this article.

The review focuses on current computational approaches to analyze peptide-protein interactions. Due to the highly flexible nature of peptides, the problem of determining the peptide-protein complex is nontrivial, and is crucial for subsequent analysis of interactions. As such, the review primarily discusses modeling peptide-protein complexes, starting with docking-based methods, continuing with the molecular-dynamics based refinements and integrative modeling platforms, and concluding with the recent breakthroughs in machine learning-based techniques. The authors also comment - in the 'MD Section' - on using molecular dynamics, in particular enhanced sampling methods, to study binding and unbinding kinetics. Ultimately, this review gives the reader a good foundation of the current tools available to both sample bound configurations of peptides with their receptors, and to generally investigate peptide-protein interactions.

Major Comments:

The review is extensive, demonstrating the rather large breadth of resources available for modeling peptides and peptide-protein complexes. Therefore, the reader can make informed decisions on which docking programs to use, whether to use an MD-based refinement strategy, and further which force fields to use for said refinement. Additionally, the authors do not shy away from discussing the limitations of the tools presented, and difficulties in the field of peptide studies generally. Due to these successes, my comments have largely been focused on discussing clarity of the review and presentation.

This review discusses numerous tools for studying peptide-protein complexes, especially the available docking and scoring programs. As a recommendation, if possible, it may be beneficial to add tables to list the common programs, the subcategories they fall in (e.g. template-based docking tool) and where they find the greatest success - if unambiguous. This will make it easy for the reader to organize the information and to quickly identify which tools they'd like to study further. It may also allow the main text to discuss the tools in greater detail.

The 'Summary' subsection of the 'Docking Section' was especially appreciated, as it tied together the salient points of docking and scoring. That being said, the bulk of the first paragraph (lines 484 to 491) would seem better placed in the beginning of the 'Scoring' section. The 'Summary' seems more appropriately initiated by line 495.

Minor Comments / Grammar:

Some of the paragraphs of this review could use transition sentences to switch topics in a coherent fashion.

It is highly recommended that the paper be analyzed for grammatical errors and typos.

All in all, this review would certainly benefit those who'd like to know more about the available tools to analyze peptide-protein complexes, thank you for your time.

---

## [Reviewer Report]

*Comments to Author*: Reviewer #1: "Modeling peptide-protein complexes: docking, simulations, and machine learning"

This review explores three main approaches in modeling peptide-protein interactions; molecular docking, molecular dynamic simulations, and machine learning. Overall it is a comprehensive and well written review that provides prospective users with the advantages and limitations of each approach. This kind of reviews is indeed of a great importance to the filed of computational chemistry as it lowers the barrier of entry to newcomers and guide them through choosing the correct approach. I suggest considering the publication of this review after performing the following minor revisions:

1.Molecular docking section: This section is well written and the authors provided a very good literature review of protein peptide-docking tools. My sole concern is that with the large amount of information provided here, the reader may find a difficulty in extracting information easily. Adding summary tables of each section can help the reader extract information efficiently. For example, the authors can provide a table summarizing the main features of each databases and sampling method [or at least the top performing], its pros, cons, and suggested application. Based on the scientific question, the prospective user can pick a database or sampling method that matches their system characteristics and refer to the the details in the main text.

2.Molecular dynamics section:

The authors provided a summary of the evolution and limitation of the current forcefields and explained the applications of different MD simulations methods in studying protein-peptide interactions. Although the authors explained the applications of several end-point free energy methods in protein-ligand/protein- peptide interactions such as MM-PBSA and MM-GBSA,they did not mention examples of using pathway free energy methods especially the highly rigorous “from statistical mechanics point of view” alchemical free energy approaches. Alchemical free energy calculations is a pathway free energy method that compute the binding free energy difference between the bound and unbound states of the peptide-protein complex by linking them through a suitable thermodynamic non physical “alchemical” path. Although the application of this approach in protein-peptide interaction is less common and more challenging than other end-point methods, it is worth shedding the light on the few successful examples of this approach as it will make this review more comprehensive [please refer to the following examples]:

1- Kilburg, D., & Gallicchio, E. (2018). Assessment of a single decoupling alchemical approach for the calculation of the absolute binding free energies of protein-peptide complexes. Frontiers in molecular biosciences, 5, 22.

2- Rashid, M. H., Heinzelmann, G., Huq, R., Tajhya, R. B., Chang, S. C., Chhabra, S.,& Kuyucak, S. (2013). A potent and selective peptide blocker of the Kv1. 3 channel: prediction from free-energy simulations and experimental confirmation.Â PloS one,Â 8(11), e78712.

3- Panel, N., Villa, F., Fuentes, E. J., & Simonson, T. (2018). Accurate PDZ/peptide binding specificity with additive and polarizable free energy simulations.Â Biophysical journal,Â 114(5), 1091-1102.

3. Machine Learning: No revision suggested.

Reviewer #2: Brief Summary:

Firstly, I would like to thank the Editorial Board of Quarterly Reviews of Biophysics, Discovery, for the opportunity to review this article.

The review focuses on current computational approaches to analyze peptide-protein interactions. Due to the highly flexible nature of peptides, the problem of determining the peptide-protein complex is nontrivial, and is crucial for subsequent analysis of interactions. As such, the review primarily discusses modeling peptide-protein complexes, starting with docking-based methods, continuing with the molecular-dynamics based refinements and integrative modeling platforms, and concluding with the recent breakthroughs in machine learning-based techniques. The authors also comment - in the 'MD Section' - on using molecular dynamics, in particular enhanced sampling methods, to study binding and unbinding kinetics. Ultimately, this review gives the reader a good foundation of the current tools available to both sample bound configurations of peptides with their receptors, and to generally investigate peptide-protein interactions.

Major Comments:

The review is extensive, demonstrating the rather large breadth of resources available for modeling peptides and peptide-protein complexes. Therefore, the reader can make informed decisions on which docking programs to use, whether to use an MD-based refinement strategy, and further which force fields to use for said refinement. Additionally, the authors do not shy away from discussing the limitations of the tools presented, and difficulties in the field of peptide studies generally. Due to these successes, my comments have largely been focused on discussing clarity of the review and presentation.

This review discusses numerous tools for studying peptide-protein complexes, especially the available docking and scoring programs. As a recommendation, if possible, it may be beneficial to add tables to list the common programs, the subcategories they fall in (e.g. template-based docking tool) and where they find the greatest success - if unambiguous. This will make it easy for the reader to organize the information and to quickly identify which tools they'd like to study further. It may also allow the main text to discuss the tools in greater detail.

The 'Summary' subsection of the 'Docking Section' was especially appreciated, as it tied together the salient points of docking and scoring. That being said, the bulk of the first paragraph (lines 484 to 491) would seem better placed in the beginning of the 'Scoring' section. The 'Summary' seems more appropriately initiated by line 495.

Minor Comments / Grammar:

Some of the paragraphs of this review could use transition sentences to switch topics in a coherent fashion.

It is highly recommended that the paper be analyzed for grammatical errors and typos.

All in all, this review would certainly benefit those who'd like to know more about the available tools to analyze peptide-protein complexes, thank you for your time.